

# Characterizing far from equilibrium states of the one-dimensional nonlinear Schrödinger equation

Abhik Kumar Saha[1,2] and Romain Dubessy[3]⋆

**1** Homer L. Dodge Department of Physics and Astronomy and Center for Quantum Research and Technology, The University of Oklahoma, Norman, Oklahoma 73019, USA
**2** School of Physical Sciences, Indian Association for the Cultivation of Science, Jadavpur, Kolkata 700032, India
**3** Université Sorbonne Paris Nord, Laboratoire de Physique des Lasers, CNRS, UMR 7538, F-93430, Villetaneuse, France

⋆ romain.dubessy@univ-paris13.fr

## Abstract

We use the mathematical toolbox of the inverse scattering transform to study quantitatively the number of solitons in far from equilibrium one-dimensional systems described by the defocusing nonlinear Schrödinger equation. We present a simple method to identify the discrete eigenvalues in the Lax spectrum and provide a extensive benchmark of its efficiency. Our method can be applied in principle to all physical systems described by the defocusing nonlinear Schrödinger equation and allows to identify the solitons velocity distribution in numerical simulations and possibly experiments.

| | |
|---|---|
| Received | 2024-05-25 |
| Accepted | 2025-02-18 |
| Published | 2025-03-06 |

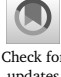
# 1 Introduction

Transport phenomena in nonlinear dispersive media may involve shock waves or the propagation of gray solitons that can be observed in a wide variety of systems. They appear for example in the propagation of monochromatic light in nonlinear optical fibers [1–5] or atomic vapors [6], ultra-cold atoms confined in very elongated traps [7–9], polariton superfluids [10] and the propagation of surface waves in deep water [11]. This can be explained by the fact that all these experiments can be modeled by a common nonlinear wave propagation equation, namely the homogeneous one-dimensional (1D) nonlinear Schrödinger equation (1DNLSE):

$$i\frac{\partial \psi(z,t)}{\partial t} = \left(-\frac{1}{2}\frac{\partial^2}{\partial z^2} + g|\psi(z,t)|^2\right)\psi(z,t), \tag{1}$$

written here in dimensionless form and where $g$ quantifies the strength of the nonlinear term and $\psi(z,t)$ is the wavefunction.

In particular equation (1) is well adapted to the description of weakly interacting 1D Bose gases held in tightly confining traps, in the mean field regime. Thanks to the fine control of ultracold quasi 1D atomic gases it is possible to study experimentally the stability of solitons [12], soliton scattering on impurities [13], soliton creation by phase imprinting [8, 14] or head-on soliton collisions [15]. This has motivated numerous theoretical works to predict the soliton dynamics, following a phase-imprinting [16, 17], in the presence of an external trap [18] or an obstacle [19, 20] and to study states with thermal-like correlations [21–24].

A wide variety of theoretical tools and analytical methods have been developed for the study of equation (1), among which the inverse scattering transform (IST) that evidenced the special role of solitons [25–29], Whitham's modulation theory to capture the transport of dispersive shock waves [30, 31], as for example in the so-called "dam-break" problem [5], and Lagrangian models describing solitons as particles with an effective mass [32–34]. When equation (1) is used to model a system with periodic boundary conditions the stationary states are known [35] and a specific form of IST can be used [36–39], taking advantage of the spatial periodicity.

More recently, following the discovery of the generalized hydrodynamic equations applied to the Lieb-Liniger model for 1D interacting bosons [40–42], the hydrodynamic approach has been adapted to the study of soliton gases, offering a new insight in the study of these systems [43–45]. The key ingredient in this approach is the knowledge of the distribution of soliton velocities [43], encoded in the discrete eigenvalues of the IST [28].

In this work we aim at contributing to the study of soliton gases by providing a accurate and robust method to identify propagating gray solitons in a far from equilibrium state evolving according to the defocusing nonlinear Schrödinger equation, Eq. (1) with $g > 0$, also known as the repulsive Gross-Pitaevskii equation. We consider a finite size system of length $L$ with periodic boundary conditions, $\psi(z,t) = \psi(z+L,t)$ and we normalize the wavefunction to the total number of particles: $N = \int_0^L dz |\psi(z,t)|^2$. In the following we denote by $n_0 = N/L$ the average density, $c = \sqrt{gn_0}$ the bare speed of sound and $\xi = 1/\sqrt{2gn_0}$ the bare healing length. These three parameters control the relevant physical scales associated to Eq. (1). We will focus on the regime of large non-linearity $gn_0 \gg 1$, such that $\xi \ll L$. To solve Eq. (1) we use a spectral method relying on a discrete grid with $N_z$ points, see Appendix A for details. In the following we will use $N_z = 512$ and a non-linearity $gn_0 = 2 \times 10^4$, such that $\delta z = L/N_z = 1.95 \times 10^{-3} < \xi = 5 \times 10^{-3}$.

Figure 1 illustrates the problem we intend to solve. Given a arbitrary initial state, propagated in time according to Eq. (1), how can we identify the number of solitons and determine precisely their velocities? This is not an easy task: even if the propagation of many density dips

**SciPost** SciPost Phys. Core 8, 028 (2025)

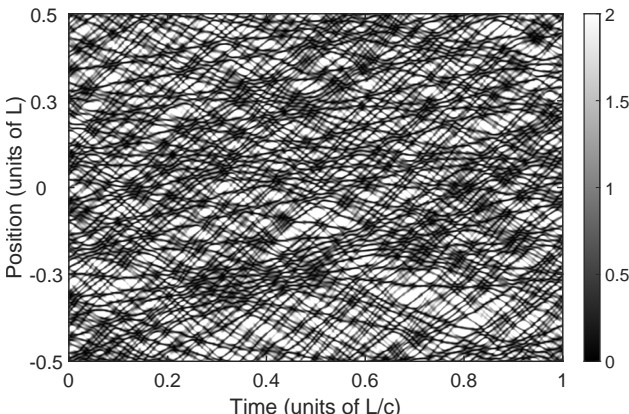

Figure 1: Density map of a far from equilibrium state evolving according to Eq. (1), exhibiting large density fluctuations, with many propagating solitons. The quantity plotted is $|\psi(z,t)|^2/N$, normalized to the number of particles. The position $z$ is normalized to the length of the system $L$ and the time to $L/c$ where $c$ is the bare speed of sound. We identify $N_s = 131.5 \pm 4.5$ solitons propagating in this particular realization. See text for details.

is clearly seen in the space-time density map of Fig. 1, it is not possible to follow accurately the trajectory of a individual soliton, due to the multiple collisions with other gray solitons.

Instead of relying on data analysis tools to solve this issue [46], our idea is to use the well established tools of the inverse scattering transform method to measure the soliton velocity distribution. More precisely we are going to use the direct scattering transform to compute the Lax spectrum and build a soliton indicator to identify the solitons in the spectrum. By doing so we are adapting to the defocusing case a method that was successful for the study of the focusing NLSE [11]. Here however we are facing two main difficulties: (a) the creation of gray solitons is a threshold-less process [9, 47], and (b) the Lax spectrum is real, requiring a careful analysis to identify solitons.

This paper is organized as follows: section 2 introduces our soliton indicator, section 3 presents a detailed benchmark of its efficiency, then section 4 discusses its applications and finally we conclude in section 5. We also provide three appendices that give additional details on the methods and a few extra examples.

## 2 Definition of a soliton indicator

The Lax operator corresponding to Eq. (1), for the defocusing case $g > 0$ reads [25, 28]:

$$\mathcal{L} = \frac{i}{2} \begin{pmatrix} \frac{\partial}{\partial z} & -\sqrt{g}\psi(z,t) \\ \sqrt{g}\psi(z,t)^* & -\frac{\partial}{\partial z} \end{pmatrix}. \tag{2}$$

Its spectrum can be computed analytically for a few simple cases, in particular for infinite size systems with well defined asymptotic densities $|\psi(z \to \pm\infty, t)| = \sqrt{n_0}$, for which it is proven that the spectrum is made of two continuous branches, separated by a gap, in which discrete eigenvalues can exist, each one corresponding to a gray soliton [25]. Since we are working with periodic boundary conditions and a finite size system, we first consider how this picture is modified. We can adapt the analytic one soliton solution of the infinite system to the case of periodic boundary conditions by considering:

$$\psi_1(z,t) = \sqrt{n_0}e^{i(k_0 z - \omega t)}\left(\cos\phi\,\tanh\left[\cos\phi\,\sqrt{gn_0}(z - \bar{z}(t))\right] + i\sin\phi\right), \tag{3}$$

where $\omega = gn_0 + k_0^2/2$ and we introduced the phase gradient $e^{ik_0 z}$, that guarantees the compatibility with periodic boundary conditions if:

$$e^{ik_0 L} = \frac{i \sin\phi - \cos\phi \tanh\left[\cos\phi \sqrt{gn_0} L/2\right]}{i \sin\phi + \cos\phi \tanh\left[\cos\phi \sqrt{gn_0} L/2\right]}. \tag{4}$$

Equation (3) describes the propagation of a single gray soliton, parametrized by the angle $\phi \in [-\pi/2, \pi/2]$, moving at a constant velocity $\dot{\bar{z}} = k_0 + c\sin\phi$. We note that Eq. (3) is only an approximate solution that is accurate only in the large non-linearity limit $L \gg \xi$. A more general solution can be found in terms of elliptic functions for arbitrary values of $L/\xi$ [35]. The angle $\phi$ sets both the speed of the soliton and the density depletion at its center, giving a relative contrast of $\cos[\phi]^2$.

Plugging the formula of Eq. (3) into the definition of Eq. (2), a lengthy but straightforward calculation shows that the Lax spectrum $\mathcal{L}v = \zeta v$ is made of two branches:

$$\zeta_q^\pm = -\frac{k_0}{4} \pm \frac{\sqrt{gn_0 + q^2}}{2}, \tag{5}$$

where $q \in 2\pi/L \times \mathbb{Z}$, corresponding to quasi plane-wave eigenvectors:

$$v_q^\pm(z,t) \propto e^{\pm iqz} \begin{pmatrix} e^{i\frac{k_0 z - \omega t}{2}}\left[i\frac{k_0 + 4\zeta_q^\pm \mp 2q}{2\sqrt{gn_0}} + \frac{\psi_1(z,t)}{\sqrt{n_0}e^{i(k_0 z - \omega t)}}\right] \\ e^{-i\frac{k_0 z - \omega t}{2}}\left[i\frac{k_0 + 4\zeta_q^\pm \pm 2q}{2\sqrt{gn_0}} - \frac{\psi_1(z,t)^*}{\sqrt{n_0}e^{-i(k_0 z - \omega t)}}\right] \end{pmatrix},$$

and a single eigenvalue:

$$\zeta_0 = -\frac{k_0}{4} - \frac{c}{2}\sin\phi, \tag{6}$$

lying in the gap between the two branches $\zeta_q^- < \zeta_0 < \zeta_q^+$ and associated to a localized eigenvector:

$$v_0(z,t) \propto \operatorname{sech}\left[\cos\phi \sqrt{gn_0}(z - \bar{z}(t))\right]\begin{pmatrix} e^{i\frac{k_0 z - \omega t}{2}} \\ -e^{-i\frac{k_0 z - \omega t}{2}} \end{pmatrix}.$$

Figure 2 shows the Lax spectrum numerically computed for the state of Eq. (3), evidencing the two branches and the isolated eigenvalue. We always sort the set of $2N_z$ eigenvalues by ascending order and focus on the central part of the spectrum, corresponding to the gap between the continuous branches, usually close to eigenvalue number $N_z$. The results are in very good agreement with the theoretical formulas presented above. The two main differences with the infinite system [25], come from the periodic boundary conditions: the Lax spectrum is globally shifted by a factor $-k_0/4$ and the two branches are made of many, closely spaced, discrete eigenvalues, because the wave-vector $q$ is discrete. The gap between the two branches is $\zeta_{q=0}^+ - \zeta_{q=0}^- = c$, which corresponds to the speed of sound set by the uniform background density, while the sum $\zeta_{q=0}^+ + \zeta_{q=0}^- = -k_0/2$ reflects the velocity of the background flow.

While it is quite obvious to identify the isolated eigenvalue in the simple Lax spectrum shown in Fig. 2, it is in general a difficult problem, as for a discrete matrix operator the spectrum is by essence discrete, and the notion of continuous branches is ill-defined. To illustrate this we consider an out-of-equilibrium state, corresponding to a small deviation with respect to a uniform background and yet resulting in a very rich dynamics, as shown on Fig. 3. We explain in Appendix A how we generate such an out-of-equilibrium state. On the one hand, the density map of Fig. 3c) shows many propagating density dips, moving at constant velocities across the sample, that can be identified as solitons by visual inspection, although it may be difficult already to count precisely the number of dips, see for example the inset of Fig. 3a). On

the other hand, the associated Lax spectrum apparently displays two branches with a single isolated eigenvalue, leading to the obviously incorrect conclusion that there is only a single soliton propagating. In fact this isolated eigenvalue corresponds to the soliton with the highest contrast (or equivalently smallest velocity). This shows the need for an unambiguous method to identify and characterize the number of solitons in an excited state of Eq. (1), for a finite-size geometry.

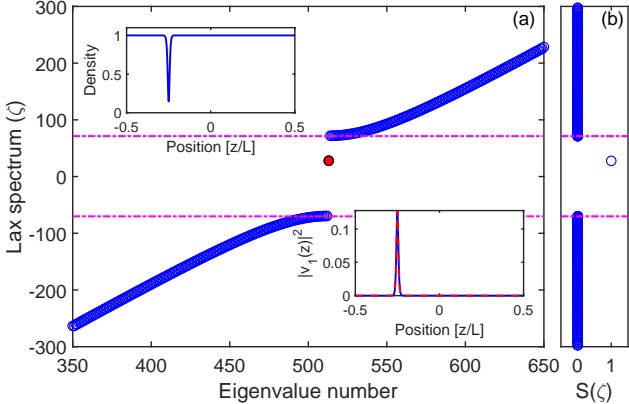

Figure 2: (color online) a) Lax spectrum for a single gray soliton state with angle $\phi = -\pi/8$: open blue circles indicate continuous branches of the spectrum, the single filled red circle is the discrete eigenvalue associated to a localized eigenstate and the two horizontal dashed magenta lines correspond to the gap boundaries at $-k_0/4 \pm c/2$. Upper inset: density of the single gray soliton state evidencing the density dip. Lower inset: square modulus of the eigenvector corresponding to the localized eigenvalue (blue solid line) compared to the analytical prediction (red dashed line). b) Soliton indicator $S(\zeta)$ for each eigenvalue, computed for the threshold $\epsilon_0 = \pi^2/(4L^2\sqrt{gn_0})$, see text for details. The only eigenvalue with an indicator equal to one corresponds to the gray soliton.

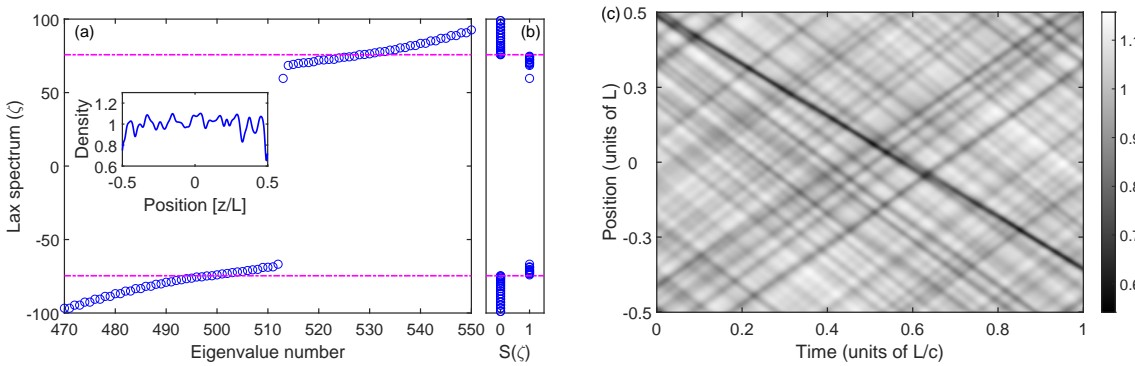

Figure 3: (color online) a) Lax spectrum for a weakly excited state (open blue circles), created using the protocol described in Appendix A. The two horizontal dashed-dotted magenta lines indicate the gap in the continuous spectrum, detected using the soliton indicator $S(\zeta)$. Inset: example of the density profile of this state (solid blue curve). b) Soliton indicator $S(\zeta)$ for each eigenvalue, computed for the threshold $\epsilon_0 = \pi^2/(4L^2\sqrt{gn_0})$, see text for details. c) Density colormap $|\psi(z,t)|^2/N$ exhibiting a small number of fast propagating solitons for this particular realization.

We intend to solve this problem by defining a soliton indicator $S(\zeta)$ equal to 1 if the eigenvalue $\zeta$ corresponds to a soliton or 0 on the contrary. To achieve this we make use of a fundamental property of integrable systems: the interaction between solitons results only in delays during their propagation, without altering their properties [28, 43]. Now taking advantage of periodic boundary conditions, consider a extended system containing two copies of the same state, as sketched on Fig. 4a). We then expect to find twice as many solitons in the extended system, with respect to the original one, each soliton being present twice, with exactly the same velocity. For the continuous branches, we expect a different behavior: the quasi plane-waves will extend over the interval of length $2L$, resulting in a denser spectrum, see for example Eq. (5).

Figure 4b) shows how the Lax eigenvalues for the initial and extended systems are distributed: the horizontal axis is shifted and rescaled to emphasize the fact that each eigenvalues corresponding to a soliton is doubly degenerate in the extended set. In the continuous branches the analysis is less obvious as the spectrum is denser.

Based on this physical intuition we build a soliton indicator by comparing the degeneracy of each eigenvalue of the Lax spectrum in the initial and extended systems: a eigenvalue with a degeneracy increased by a factor of two will correspond to a soliton. As we will show below, this allows to identify properly the propagating solitons in a arbitrary out-of equilibrium state of Eq. (1). To implement this, we compute the set of eigenvalues $\{\zeta_i\}$ of the initial system and the set of eigenvalues $\{\zeta'_i\}$ of the extended system. We then define for each eigenvalue $\zeta_i$ the soliton indicator as:

$$S(\zeta_i) = \frac{\text{card}\{\zeta'_j \text{ such that } |\zeta_i - \zeta'_j| < \epsilon\}}{\text{card}\{\zeta_j \text{ such that } |\zeta_i - \zeta_j| < \epsilon\}} - 1, \tag{7}$$

which is the ratio of the number of eigenvalues close to $\zeta_i$ in the extended set and original set, respectively, minus one. We use here the card notation to denote the cardinality of a set of eigenvalues. If the degeneracy of eigenvalue $\zeta_i$ is doubled in the extended set, we find $S(\zeta_i) = 1$, while if the degeneracy does not change $S(\zeta_i) = 0$. In the following we define the number of solitons in a particular state $N_{\text{sol}}$ as the number of eigenvalues with indicator equal to one.

Equation (7) introduces a parameter $\epsilon$ that defines the threshold between degenerate and non-degenerate eigenvalues. Because we have in mind to apply this method to numerical simulations with finite grid size and discrete approximations of the operators, therefore introducing some level of numerical error, we have to choose a finite threshold value. Looking at the analytical formula corresponding to the single soliton solution, we may use Eq. (5) to estimate the minimal distance between two eigenvalues at the gap edge ($q \to 0$): $\delta\zeta \simeq \delta q^2/(4\sqrt{gn_0}) = \pi^2/(L^2\sqrt{gn_0})$, for the initial system, where $\delta q = 2\pi/L$. In the extended system this value is divided by a factor of 4 and it seems then reasonable to choose a value $\epsilon \leq \epsilon_0 = \pi^2/(4L^2\sqrt{gn_0})$ to separate two distinct eigenvalues in the extended set.

Figure 4 illustrates this concept of soliton indicator, using a artificially created (see section 3) state for which solitons can be clearly identified as density dips by visual inspection, see the initial density profile shown in Fig. 4a) and the density map of Fig. 4d). For this particular realization the soliton indicator, computed for the threshold value $\epsilon_0$, identifies 10 eigenvalues with $S(\zeta) = 1$, corresponding to the 10 imprinted solitons. Similarly, Fig. 2b) and 3b) show the value of $S(\zeta)$, computed with the same value $\epsilon_0$, for a simple state with a single soliton and a out-of-equilibrium state, respectively. It seems to behave as anticipated, identifying a few eigenvalues as solitons inside the gap between the two continuous branches. In section 3 we benchmark how the soliton indicator behaves with the choice of threshold and give a robust definition of the number of solitons.

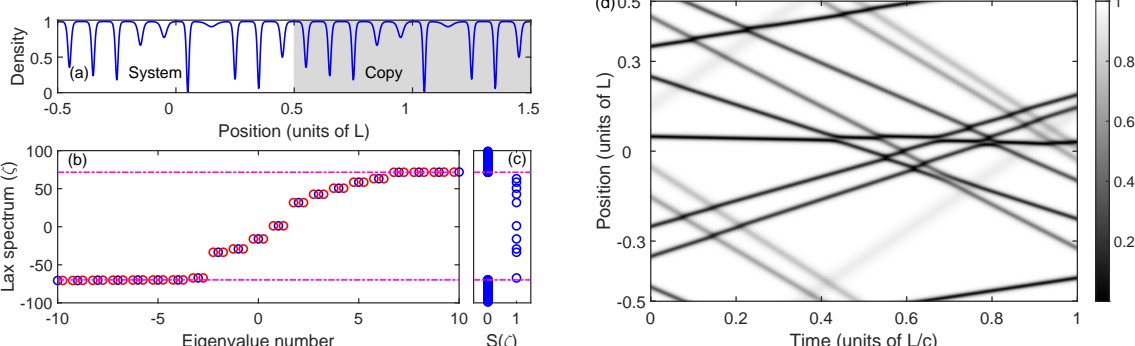

Figure 4: (color online) a) Density as a function of position for a state with $N_{\text{imp}} = 10$ solitons using the analytical formula described in Eq. (8). The solitons can be clearly identified as density dips by visual inspection. The gray shaded area represents the system copy we use to define the extended system. b) Lax spectrum for the initial (open blue circles) and extended (open red circles) systems, respectively. The two dashed-dotted magenta lines indicate the gap in the continuous part of the spectrum detected using the soliton indicator $S(\zeta)$. c) Soliton indicator $S(\zeta)$ for each eigenvalue, computed for the threshold $\epsilon_0$. d) Density colormap $|\psi(z, t)|^2/N$ of the propagating solitons for this particular realization.

Interestingly we also find with the indicator the two gap edges $\zeta^{\pm}_{q=0}$, corresponding to the last (first) eigenvalue of the quasi continuous lower (upper) branch. In analogy with the analytical formula of the single soliton solution, see above, we may interpret the amplitude of the gap $\zeta^{+}_{q=0} - \zeta^{-}_{q=0} = c_{\text{eff}}$ as the effective speed of sound in the sample and deduce the effective background flow: $k_{\text{eff}} = -2(\zeta^{+}_{q=0} + \zeta^{-}_{q=0})$.

## 3 Benchmark of the soliton indicator

In this section we study systematically how the choice of the threshold $\epsilon$ in the soliton indicator of Eq. (7) affects the efficiency of the soliton detection. To do so, we study the Lax spectrum of states defined by the following analytical formula:

$$\psi_M(z) = \sqrt{n_0} \prod_{j=1}^{M} e^{ik_j z} \left( \cos\phi_j \tanh\left[ \cos\phi_j \sqrt{gn_0}(z - z_j) \right] + i \sin\phi_j \right), \tag{8}$$

where $M > 0$ is an integer. For $M = 1$, Eq. (8) is simply Eq. (3) evaluated at $t = 0$, thus describing a single gray soliton with angle $\phi_1$ located at position $z_1$, provided that $k_1$ is chosen according to Eq. (4) to fulfill periodic boundary conditions. For $M > 1$ it is a reasonable assumption that Eq. (8) describes a state with $M$ solitons, provided that they are initially located far from each other. In other words, we use Eq. (8) as a guess state containing a dilute gas of $M$ gray solitons that we will use to benchmark our soliton detection method. We will proceed as follows. First we choose a value of $M \geq 1$, then we draw randomly $M$ phases $\phi_j \in [-\pi/2, \pi/2]$ and build the state of Eq. (8) by distributing the $M$ solitons at regular spacings $z_{j+1} - z_j = L/M$ over the interval $[0, L]$. We then compute the Lax spectrum for this state and study how the soliton indicator changes with the choice of threshold $\epsilon$. We repeat this procedure many times to sample the different possible soliton gas configurations and analyze the results to define a probability of success for our soliton detection method.

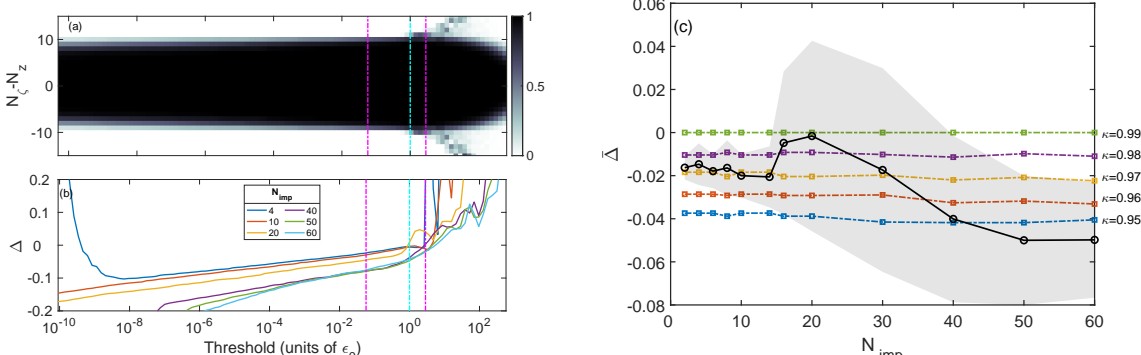

Figure 5: (color online) a) Average value of the soliton indicator over 500 realizations for $N_{\text{imp}} = 20$, corresponding to 10 imprinted pairs of solitons as a function of the threshold in units of $\epsilon_0$ and the index of the eigenvalue in the spectrum $N_\zeta$. We perform the numerical simulation with grid size $N_z = 1024$ and nonlinearity $gn_0 = 2 \times 10^4$. The two vertical dashed-dotted magenta lines corresponds to the two thresholds $\epsilon_- = 0.05\epsilon_0$ and $\epsilon_+ = 2.86\epsilon_0$. The cyan vertical dash-dotted line corresponds to the threshold value $\epsilon_0$. b) The relative difference between the average number of detected and imprinted solitons $\Delta = (N_{\text{sol}} - N_{\text{imp}})/N_{\text{imp}}$ as a function of thresholds for small to large number of imprinting solitons. For each curves, the results are averaged over an ensemble sampling at least 10000 random phases. c) The relative difference between the average number of detected and imprinted solitons $\bar{\Delta} = (\bar{N}_{\text{sol}} - N_{\text{imp}})/N_{\text{imp}}$ (black open markers with solid black line) with average uncertainty represented by the gray shaded area as a function of the number of imprinted solitons $N_{\text{imp}}$. The different dashed-dotted lines with square markers represent the value of $\Delta$ as a function of $N_{\text{imp}}$ with phases below a certain value, $|\phi| < \kappa\pi/2$. Here we only plot five different $\kappa$ values ranging from $\kappa = 0.95$ to $\kappa = 0.99$.

As mentioned in the previous section, an annoying consequence of working with periodic boundary conditions is the extra phase gradient proportional to $k_j$ that must be included in the definition of the state and that results in a shift of the gap edges. One way to avoid this issue is to consider a special case with a gas of pairs of solitons having opposite phases: all phase gradients then compensate and the Lax spectrum becomes symmetric, all eigenvalues coming by pairs of opposite sign. We will first discuss the case of a dilute gas of soliton pairs before generalizing to a dilute gas of solitons. In the following, $N_{\text{imp}} = M$ will be the number of solitons we intend to imprint, using Eq. (8) and $N_{\text{sol}}$ the number of solitons detected for a particular threshold and realization.

## 3.1 Dilute gas of soliton pairs

A first generic feature that we observe is that $S(\zeta)$ is always 0, except for a small number of eigenvalues, in the central part of the spectrum (assuming that the eigenvalues are sorted by increasing value). The number of eigenvalues corresponding to $S(\zeta) = 1$, that we denote $N_{\text{sol}}$ is always smaller than $N_{\text{imp}}$ for small thresholds and tends to increase with the value of the threshold.

Figure 5a) illustrates this by showing the value of $S(\zeta)$ for $N_{\text{imp}} = 20$, corresponding to 10 imprinted pairs, averaged over 500 realizations with phases $\phi_j$ drawn randomly, as a function of the threshold $\epsilon$ and the index of the eigenvalue in the spectrum $N_\zeta$. Figure 5b) shows the relative difference between the average number of detected and imprinted solitons $\Delta = (N_{\text{sol}} - N_{\text{imp}})/N_{\text{imp}}$, as a function of the threshold and for initial states with different num-

bers of pairs. For each curves the results are averaged over an ensemble sampling at least 10000 random phases. From Figure 5b) we see that a low threshold value leads to an underestimation of the number of solitons, while a high threshold value induces an overestimation. The analytical threshold value $\epsilon_0$ derived in Section 2 tends to underestimate the number of solitons for high soliton density.

In order to provide a reasonable estimate of the number of solitons in a single realization we define empirically two thresholds: $\epsilon_- = 0.05\epsilon_0$ and $\epsilon_+ = 2.86\epsilon_0$, that give a lower ($N_{\mathrm{sol}}^-$) and upper ($N_{\mathrm{sol}}^+$) bound respectively on the number of solitons. We have chosen those particular values such that the average magnitude of $\Delta$ remains below 0.1, corresponding to an error of less than 10%, on average. This allows us to measure the number of soliton *in a single realization* $N_{\mathrm{sol}} = \bar{N}_{\mathrm{sol}} \pm \delta N_{\mathrm{sol}}$, where $\bar{N}_{\mathrm{sol}} = (N_{\mathrm{sol}}^+ + N_{\mathrm{sol}}^-)/2$ and $\delta N_{\mathrm{sol}} = (N_{\mathrm{sol}}^+ - N_{\mathrm{sol}}^-)/2$.

Figure 5c) reports the average value of the relative number of detected solitons with the average uncertainty, as a function of the number of imprinted solitons. It shows that with our choice of thresholds $\epsilon_{\pm}$, $\bar{N}_{\mathrm{sol}}$ tends to underestimate the number of imprinted solitons by 2 to 5%. We also report the relative number of imprinted solitons with phases below a certain value, in magnitude: $|\phi| < \kappa\pi/2$, with $\kappa \in [0, 1]$. We find that our method gives a average number of solitons that agrees very well with the number of imprinted solitons with phases below $0.97 \times \pi/2$, within self-consistently estimated uncertainties. The agreement is less good for higher numbers of pairs, which we attribute to the fact that the analytical formula is not accurate anymore because the solitons tend to overlap at higher density. However we think that our method can still faithfully estimate the number of solitons in the sample with a reasonable uncertainty. We note that solitons with phases greater than $0.97 \times \pi/2$ correspond to dips in the density profile with contrast below $3 \times 10^{-3}$, using a naive estimation based on Eq. (3), that are hardly visible anyway. We stress also that the results presented in Fig. 5 are ensemble averaged, in order to calibrate the relevant thresholds and the efficiency of the protocol. However when applied to particular states our method can identify exactly the number of solitons, when the contrast of solitons is sufficiently high, even if they partially overlap.

So far we have only considered the number of detected solitons, we now study how the eigenvalues are distributed. To do so we again use the state of Eq. (8) and identify for each realization of the random phases the eigenvalues corresponding to solitons. More precisely we build a first histogram of the eigenvalues identified by the lower threshold $\epsilon_-$ and a second one for the extra ones identified only by the second threshold $\epsilon_+$. Figure 6a) shows the result for 5 pairs, averaged over 1000 realizations, and compare it to the expected histogram, obtained from the knowledge of all the phases and using Eq. (6) with the bare speed of sound. The measured histogram shows a very good agreement with the expected one, except near the edges (see the inset), as could be anticipated. The dash-dotted cyan vertical line indicates the eigenvalues corresponding to $\phi = \pm 0.97 \times \pi/2$: it corresponds to the peak of the $\epsilon_+$ histogram and confirms the analysis of Figure 5c): only the fastest solitons with phases $|\phi| > 0.97 \times \pi/2$ are not detected.

We have varied the number of imprinted pairs between 1 and 30, always sampling roughly 10000 phases (in total) and observed qualitatively a similar behavior. However for number of pairs larger than 10 we start to observe significant deviations. To confirm this we extract from each realization an estimation of the speed of sound $\bar{c}_{\mathrm{eff}} \pm \delta c_{\mathrm{eff}}$ (from the gap edges corresponding to the two thresholds, see Sec. 2) and plot the averaged effective speed of sound and uncertainty as a function of the number of pairs on Fig. 6b). We attribute the fact that the speed of sound deviates from the background value $c = \sqrt{gn_0}$ to the failure of Eq. (8) to describe a dense soliton gas. However we think that the method we explained here still gives a reasonable estimate of the soliton gas properties.

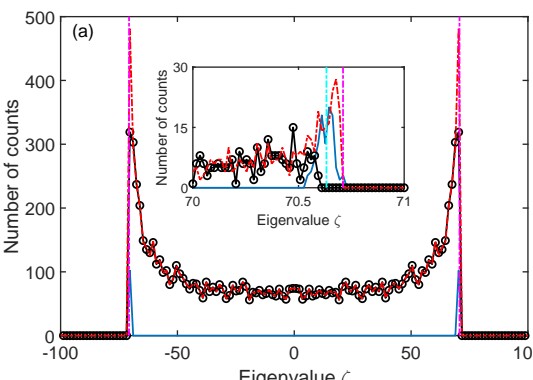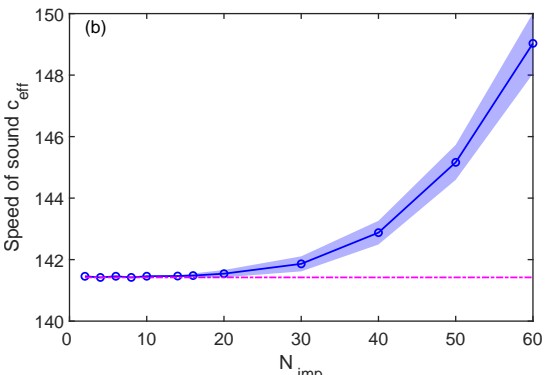

Figure 6: (color online) a) Histogram of soliton eigenvalues, computed for a state with 5 imprinted pairs ($N_{\mathrm{imp}} = 10$), averaged over 1000 realizations. The black solid line with open circles shows the histogram computed for eigenvalues identified by the threshold $\epsilon_-$, the blue solid line shows the histogram for extra eigenvalues identified only by the threshold $\epsilon_+$ and the dashed-dotted red line the expected distribution of imprinted solitons. The inset evidences the behavior near the gap edge where the solitons are more difficult to detect. The dash-dotted vertical magenta lines correspond to $\pm c/2$, while the dash-dotted cyan lines correspond to $\pm \sin(0.97\pi/2)c/2$. b) Effective speed of sound $c_{\mathrm{eff}}$ (open blue circles) found from the gap edges as a function of the number of imprinted solitons. The solid blue line is a guide to the eye. The light blue shaded area indicates the uncertainty on the speed of sound, extracted from the gap edges computed with the thresholds $\epsilon_\pm$, see text for details. The horizontal dash-dotted magenta line indicates the bare speed of sound $c$.

## 3.2 Dilute gas of solitons

We now briefly confirm our choice of thresholds by studying a dilute gas of solitons, each phase being drawn independently. Figure 7a) reports the average relative error on the number of solitons that remains consistent with the results of Fig. 6b) (note that 30 solitons is equivalent to 15 pairs): on average solitons with phases of magnitude up to $0.97 \times \pi/2$ are correctly detected. Figure 7b) evidences that the distribution of detected eigenvalues indeed follows the expected law. However, one notable difference is the distribution of extra eigenvalues (those detected only in between $\epsilon_-$ and $\epsilon_+$), near the edges of the histogram that is much broader. This is not due to a lesser precision of the detection method, but reflects the random shifts of the gap edges in the continuous spectrum due to the background phase, as mentioned above (see Eqs. (5), (6) and (8)).

For example, when applied to very simple states, as shown on Fig. 2 for a single soliton or on Fig. 4 for a state with 10 clearly visible solitons, our soliton detection method gives an exact result: $N_{\mathrm{sol}} = 1 \pm 0$ or $10 \pm 0$, respectively, which can be simply confirmed in that case by a visual inspection of the space-time density map. Finally our method enables a simple and efficient identification of solitons for arbitrary excited states, as shown in Fig. 1 and 3, resulting in $N_{\mathrm{sol}} = 131.5 \pm 4.5$ and $25.5 \pm 2.5$ respectively, with a guarantee that the uncertainty concerns only the shallowest (fastest) solitons. Moreover we emphasize that our method give also access to the full distribution of Lax eigenvalues corresponding to the soliton gas, a key ingredient in the generalized hydrodynamics approach [43].

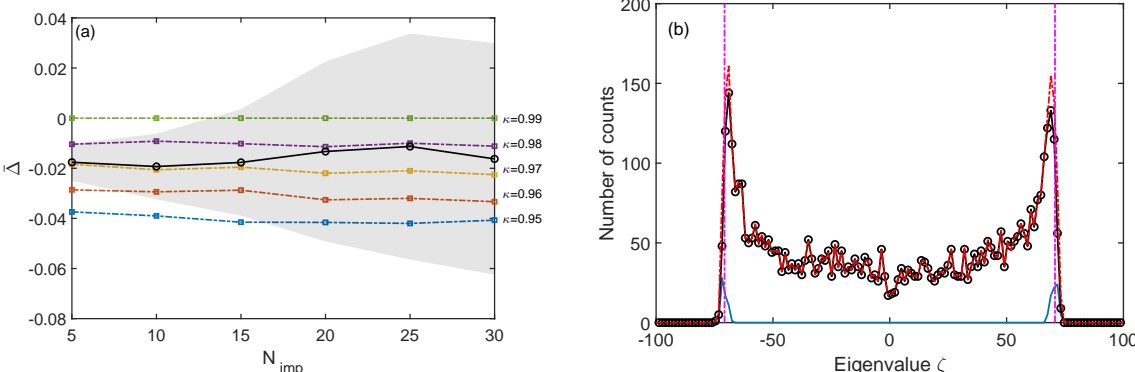

Figure 7: (color online) a) The relative difference between the average number of detected and imprinted solitons $\bar{\Delta}$ (black open markers with solid black line) with average uncertainty represented by the gray shaded area as a function of the number of imprinted solitons $N_{\mathrm{imp}}$. The different dashed-dotted lines with square markers represent the value of $\Delta$ as a function of $N_{\mathrm{imp}}$ with phases below a certain value, $|\phi| < \kappa \pi / 2$. Here we only plot five different $\kappa$ values ranging from $\kappa = 0.95$ to $\kappa = 0.99$. b) Histogram of soliton eigenvalues, computed for a state with 10 imprinted solitons $N_{\mathrm{imp}} = 10$, averaged over 500 realizations. The black solid line with open circles shows the histogram computed for eigenvalues identified by the threshold $\epsilon_-$, the blue solid line shows the histogram for extra eigenvalues identified only by the threshold $\epsilon_+$ and the dashed-dotted red line the expected distribution of imprinted solitons. The dash-dotted vertical magenta lines correspond to $\pm c/2$.

## 3.3 Comparison to a analytical result

Finally to validate our findings we compare the outcome of our method with a simple analytical result. A initial state with a hyperbolic tangent wavefunction is known to generate a odd number of solitons with a specific distribution of velocities in a infinite size system [9]. We can adapt this result to the case of periodic boundary conditions, by using the initial state:

$$\psi_{\mathrm{ana}}(z) = \sqrt{n_0} \tanh(\sqrt{g n_0} \alpha z) e^{i k_0 z} \,, \tag{9}$$

where the phase gradient $e^{i k_0 z}$ is there to compensate the phase jump at $z = 0$ and ensure compatibility with periodic boundary conditions. The parameter $\alpha$ controls the width of the initial density dip and the distribution of generated solitons. Using this initial wavefunction, the Lax operator of Eq. (2) can be diagonalized exactly and the resulting eigenvalues of the discrete spectrum are given by $\zeta_0 = -\frac{k_0}{4}$, $\zeta_{2j} = -\frac{k_0}{4} + \frac{c}{2}\sqrt{1 - (1 - j\alpha)^2}$, $\zeta_{2j+1} = -\frac{k_0}{4} - \frac{c}{2}\sqrt{1 - (1 - j\alpha)^2}$ where $j = 1, 2, ..., N_0$ and $N_0$ is the largest integer such that $N_0 < 1/\alpha$. These formula show that for arbitrary $\alpha$, the initial wavefunction profile of the form of Eq. (9) always produces a dark soliton at $z = 0$ and additional $N_0$ pairs of symmetric gray solitons corresponding to the nonzero eigenvalues. As a result, the total number of eigenvalues and thus the total number of solitons is $2N_0 + 1$ and depends on the value of $\alpha$. We now benchmark our soliton detection method using Eq. 9 for different values of $\alpha$, using the lower threshold $\epsilon_- = 0.05\epsilon_0$ and upper threshold $\epsilon_+ = 2.86\epsilon_0$.

Figure 8a) shows a typical Lax spectrum we obtain for the state of Eq. (9), here for the particular value $\alpha = 0.26$. Our soliton indicator method identifies correctly the gap between the two continuous branches, and $N_{\mathrm{sol}} = 8 \pm 1$ solitons. The analytical formula predicts exactly 7 solitons. By comparing the predicted eigenvalues to the detected ones, we find a excellent agreement: all expected solitons are correctly detected and the two extra ones correspond to very shallow features close to the gap. This result is consistent with our analysis of the dilute

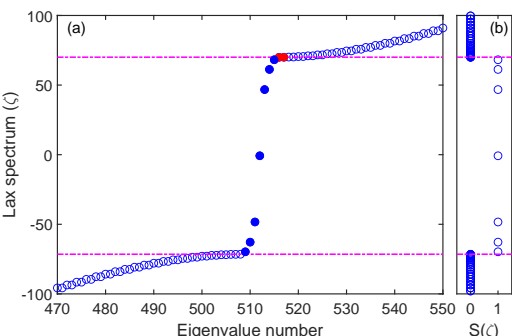 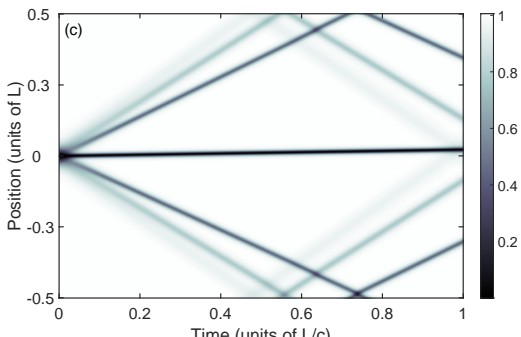

Figure 8: (color online) a) Lax spectrum as a function of the eigenvalue number for $\alpha = 0.26$ (open blue circles) using the initial state of Eq. (9). The two horizontal dashed-dotted magenta lines indicate the gap in the continuous spectrum, detected using the soliton indicator $S(\zeta)$. The filled blue circles denote the eigenvalues corresponding to the $\epsilon_-$ threshold, while the two filled red circles denote the extra eigenvalues detected only by the $\epsilon_+$ threshold. b) Soliton indicator $S(\zeta)$ for each eigenvalue, computed for the threshold $\epsilon_-$. c) Density colormap showing the propagation of the initial state, with seven gray solitons propagating.

soliton gas. In Fig. 8c), we plot the density colormap corresponding to the propagation of the initial state. Here it can be checked that indeed seven density dips (gray lines) are propagating, although the fastest (shallowest) ones are barely visible on this color scale. It is interesting to note also that the solitons moving upwards (relative to the orientation of the plot) are faster than those moving downwards: the symmetry is broken by the background phase gradient $e^{i k_0 z}$.

In order to evaluate the accuracy of our soliton indicator we repeat the same study for 200 different values of $\alpha \in [0.05, 0.90]$. Fig. 9 reports the result of this analysis, in the form of the histogram of eigenvalues corresponding to solitons. We compare this histogram to the one we expect based on the analytical formulas. We find an excellent agreement, except near the edges where our choice of upper threshold tends to lead to "false positive" soliton detection, as was observed for the previous study. Here we limit ourselves to $\alpha > 0.05$ to avoid distortion of the hyperbolic tangent shape due to periodic boundary conditions: when $\alpha < 0.05$ the effective speed of sound we measure deviates significantly from the bare value $c$, which provides an indication that the background is not flat anymore.

To summarize, we have shown in section 3 that it is possible to use the soliton indicator of Eq. (7) with two thresholds to estimate a upper and a lower bound of the soliton number, the effective speed of sound with an uncertainty and finally the distribution of soliton velocities. By using the two benchmarking methods we demonstrate that we are able to identify all solitons and that when extra solitons are detected it is always near the gap edges, which leads only to a small overestimation of the speed of sound. Importantly our method can then be applied to identify solitons in a arbitrary initial state.

# 4 Discussion

In this last section we discuss our results and highlight a few open questions.

We have checked how the choice of the thresholds $\epsilon_\pm$ was related to the value of the nonlinearity. To do so, we have repeated the benchmarking procedure for different values of the nonlinearity $g n_0$ and we have found that choosing the thresholds as $\epsilon_\pm \to \epsilon_\pm \times \sqrt{2 \times 10^4 / (g n_0)}$ provide similar performances, keeping in mind that our benchmarking protocol using the for-

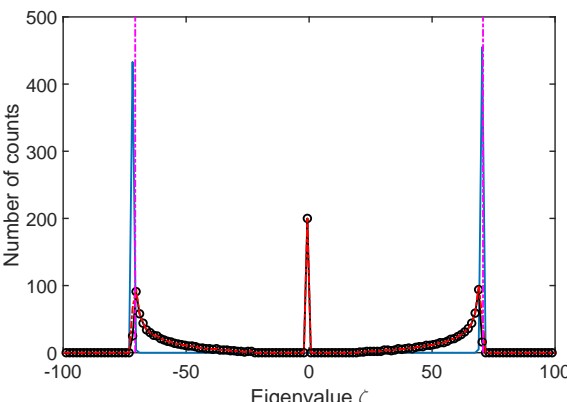

Figure 9: (color online) Histogram of the soliton eigenvalues computed for a range of $\alpha$ values. The black solid line with open circles shows the histogram computed for eigenvalues identified by the threshold $\epsilon_-$, and the blue solid line is the histogram for extra eigenvalues identified only by the threshold $\epsilon_+$. The red dashed line shows the histogram computed for eigenvalues for the soliton as described in Eq. (9). The two dashed-dotted vertical magenta lines indicate the gap in the Lax spectrum at the positions $\pm c/2$.

mula of Eq. (8) requires that $L \gg \xi$ (or $gn_0 \gg 1$). Similarly, we have also checked that our method is robust with respect to the grid size, provided that $\delta z = L/N_z < \xi$, by testing different values of $N_z \in \{256, 512, 1024\}$. Even for $N_z = 256$ (meaning that $\delta z \simeq \xi$) we find that the soliton indicator correctly identifies the discrete eigenvalues in a dilute soliton gas initial state, see Eq. (8). However, we note that with this lower grid size the time propagation induces numerical errors and that the Lax spectrum evolve slightly with time, indicating that the discretization is too sparse. This is in fact a very convenient way to check whether the time evolution of Eq. (1) is computed with a sufficiently high accuracy. Indeed, if the continuous model is correctly mapped onto a discrete grid, we find that it remains approximately integrable, in the sense that the Lax spectrum is indeed time-independent. However, it may not be the case, for example if the number of modes is too small ($\delta z \geq \xi$) or if the time propagation introduces errors.

While varying the grid size $N_z$ or the nonlinearity $gn_0$ we noticed that for some trial functions the soliton indicator could take values other than 0 or 1, depending on the threshold value $\epsilon$. This always occurs near the edges of the continuous branches, where, as discussed above it is difficult to distinguish very shallow solitons. For the purpose of our analysis we have considered that these few eigenvalues with "anomalous" indicator (meaning $S(\zeta) \neq 0$ or 1) belonged to the same class as their neighboring eigenvalues. We interpret this phenomenon as a consequence of the fact that gray solitons are created without threshold in the defocusing 1DNLSE.

We emphasize that our method gives access to an estimation of the effective speed of sound in the system (see the end of section 2), which can be much larger than the bare estimate $c = \sqrt{gn_0}$. To the best of our knowledge there is no other method to access directly this quantity for a arbitrary initial state. For example, in Fig. 1 it can be seen that some propagating features travel much faster than the bare speed of sound, and their speed is compatible with the measured speed of sound $c_{\text{eff}} = 268 \pm 4$ (in dimensionless units). Moreover we obtain the whole distribution of soliton velocities and it would be very interesting to apply our method to a set of thermal states, generated by a stochastic classical field model [24]. In principle it should be possible to obtain the equation of state $c_{\text{eff}}(\mu, T)$, relating the speed of sound to the chemical potential $\mu$ and the temperature $T$, and compare it to analytical predictions [48].

Similarly, we may use our soliton indicator tool to test the accuracy of the recently proposed generalized hydrodynamics equation for gray soliton gases [43]. Indeed, once the distribution of Lax eigenvalues corresponding to solitons is known, we may select a particular one as a test "tracer" soliton and follow its trajectory over time. To do so we may look at the position of the maximum of the associated eigenvector, see the lower inset of Fig. 2a). By comparing the initial and final position associated to the same eigenvalue (recall that the Lax spectrum is time invariant) we obtain a direct measure of the effective velocity of the soliton, that can be compared to the hydrodynamic effective velocity [43].

A natural extension of our work would be to find a way to extend our soliton detection method to non-integrable cases, that are highly relevant in the context of atomtronics: for example introduce a localized barrier that will change the soliton velocities [49, 50] or enable the nucleation of new solitons [19, 20]. In the limit of large barriers, that is considering hard wall boundary conditions the Lax spectrum can be readily computed using a mirror image technique [51]. It would be also very relevant to consider the case of a harmonic potential: although it breaks formally the integrability of Eq. (1) it sustains long-lived soliton like solutions [9]. For all these examples we believe that our method can be adapted to study quantitatively how the external potential term breaks integrability, in the spirit of the study reported in [11] for the focusing (attractive) case. More generally it would be interesting to extend our results to other integrable equations similar to Eq. (1) [52, 53].

Finally, we emphasize that the method we introduce in this work is rather simple as it requires only Fourier transforms and the diagonalization of a simple matrix, see Appendix B for details. Therefore it can be readily applied to any simulation relying on Eq. (1). We provide in Appendix C two examples of specific states with peculiar behavior that can be identified only with the use of the soliton indicator. We also think that it can even be used to analyze experimental data, provided that one can measure the amplitude and the phase of the field $\psi(z, t)$ at a given time, with a spatial resolution of at least $\xi$, and calibrate the non-linearity parameter $g$. This is within reach in experiments dealing with the propagation of light pulses in nonlinear fibers [54] or atomic vapors [6].

## 5   Conclusion

We have reported a detailed study of the use of the inverse scattering transform tools to identify the number of gray solitons in the defocusing one-dimensional nonlinear Schrödinger equation. We define a self consistent soliton indicator that allows a study of the soliton distribution and demonstrate through a extensive benchmark its reliability. More precisely we are able to count the solitons and find their velocities within a given margin of error, where the errors always concern fast, shallow, gray solitons. Moreover it provides a accurate measurement of the effective speed of sound in a arbitrary excited state. We think that our method is very relevant to the analysis of gray soliton gases, in the context of a generalized hydrodynamics approach.

## Acknowledgments

RD thanks Maxim Olshanii for introducing him to the inverse scattering transform method and for many inspiring discussions. LPL is UMR 7538 of CNRS and Sorbonne Paris Nord University.

**Funding information**   RD would like to thank the Institut Henri Poincaré (UAR 839 CNRS-Sorbonne Université) and the LabEx CARMIN (ANR-10-LABX-59-01) for their support.

# A Methods

We provide here details on the numerical methods we use to study the 1DNLSE. To numerically solve Eq. (1), we use a spectral method relying on fast Fourier transforms to evaluate exactly the kinetic energy term [55], with a regular grid of $N_z = 512$ points and a dimensionless nonlinear parameter $gN = 2 \times 10^4$, well within the mean field regime [50]. The grid introduces a natural cut-off for the wavevectors at $k_{max} = \pi N_z / L$ and to avoid aliasing in the computation of nonlinear terms we use a projector onto the low $k$ region: $|k| < k_{cut} = 2k_{max}/3$ [56].

The groundstate of Eq. (1) corresponds to a flat density profile: $\psi_0 = \sqrt{n_0}$, where $n_0 = N/L$, fixing the value of the bare speed of sound $c = \sqrt{gn_0}$ and healing length $\xi = 1/\sqrt{2gn_0}$.

To drive the system to out-of-equilibrium states we use a simple excitation protocol: we introduce a Gaussian potential that we stir back and forth along the $z$ axis [57–61]. To do so, we add to Eq. (1) the excitation potential in the form of a moving Gaussian obstacle:

$$V_{stirr}(t, z) = V_b(t) \exp[-(z - z_c(t))^2 / \sigma^2],$$

where $V_b(t)$ is the time-dependent barrier height, $\sigma = 4\xi$ is the width of the barrier and $z_c(t) = \delta z \cos(\omega_{exc} t)$ is the position of the barrier. The amplitude of the motion is set to $\delta z = L/4$, the barrier is turned on in a time $t_{on} = L/c$, kept at its maximum value $V_0 = gn_0$ for $T_{exc} = 8 \times L/c$ and turned off in a time $t_{off} = L/c$. $V_0$ is chosen such that the density is nearly completely depleted at the peak of the barrier which facilitates the creation of solitons, while $t_{on}$ and $t_{off}$ are slow enough to prevent the creation of excitations if the barrier is not moving (i.e. for $\omega_{exc} = 0$ or $\delta z = 0$). We then vary $\omega_{exc}$ to control the amount of excitation created in the final state. For example the state of Fig. 1 was generated with a fast oscillation $\omega_{exc} = 4.5 \times c/L$, while the state of Fig. 3 corresponds to a slower frequency $\omega_{exc} = 0.384 \times c/L$. At the end of the excitation phase, when the barrier amplitude is turned off, we record the wavefunction and compute its Lax spectrum. Our analysis protocol allows then to extract for each simulation the number of solitons.

# B Computation of the Lax spectrum

The Lax eigenvalue equation we want to solve is $\mathcal{L}v = \zeta v$, with $v = (v_1, v_2)^T$ a two-component vector. In order to compute this equation in momentum space, we take the Fourier transform, $\hat{\mathcal{L}} * \hat{v} = \zeta \hat{v}$, where $*$ is the convolution and $\hat{v}$ is the Fourier transform of $v$. The equation then reads:

$$\begin{pmatrix} -\frac{k}{2}\hat{v}_1 - i\frac{\sqrt{g}}{2}\hat{\psi} * \hat{v}_2 \\ i\frac{\sqrt{g}}{2}\hat{\psi}^* * \hat{v}_1 + \frac{k}{2}\hat{v}_2 \end{pmatrix} = \zeta \begin{pmatrix} \hat{v}_1 \\ \hat{v}_2 \end{pmatrix}.$$

Now to compute the convolution, corresponding to off-diagonal terms in $\mathcal{L}$, we have to write it using the discrete Fourier transform: $\hat{\psi}(k) \rightarrow \hat{\psi}_q = \sum_{n=0}^{N_z-1} \psi_n e^{-ik_q x_n}$ where $x_n$ or $k_q$ belong to the discrete grid in position or momentum space. Then the discrete convolution reads:

$$[\hat{\psi} * \hat{v}]_q = \sum_{p=0}^{N_z-1} \hat{\psi}_{q-p} \hat{v}_p,$$

where the index $q - p$ in the sum is taken modulo $N_z$, as $\hat{\psi}_q = \hat{\psi}_{q+N_z}$. This operation can be written in a matrix form:

$$C\hat{v} = \begin{pmatrix} \hat{\psi}_0 & \hat{\psi}_1 & \cdots & \hat{\psi}_{N_z-2} & \hat{\psi}_{N_z-1} \\ \hat{\psi}_{N_z-1} & \hat{\psi}_0 & \cdots & \hat{\psi}_{N_z-3} & \hat{\psi}_{N_z-2} \\ & & \cdots & & \\ \hat{\psi}_2 & \hat{\psi}_3 & \cdots & \hat{\psi}_0 & \hat{\psi}_1 \\ \hat{\psi}_1 & \hat{\psi}_2 & \cdots & \hat{\psi}_{N_z-1} & \hat{\psi}_0 \end{pmatrix} \hat{v},$$

from which it is clear that the convolution matrix has a Toeplitz structure.

Finally we diagonalize the $2N_z \times 2N_z$ matrix:

$$\hat{L} = \frac{i}{2} \begin{pmatrix} ik & -\sqrt{g}C \\ \sqrt{g}C^\dagger & -ik \end{pmatrix},$$

where $k$ stands for the diagonal matrix of discrete wave-vectors. All operations are implemented in Octave/Matlab language using built-in functions.

As $\hat{L}$ is a generic hermitian matrix, the typical algorithmic complexity cost of its diagonalization is $\mathcal{O}(N_z^3)$ [62]. For the grid sizes we used in this work we have not found any significant difference in computation time between computing only the eigenvalues or the eigenvalues and the eigenvectors. However the latter requires more available memory to store the matrix of eigenvectors. Our soliton detection algorithm requires to diagonalize first a $2N_z \times 2N_z$ matrix and then a $4N_z \times 4N_z$ matrix, and to identify the soliton positions we need the eigenvectors of the first matrix. In our Matlab implementation we indeed observe that the diagonalization of the second matrix takes roughly 8 times longer than the first one. For a grid size of $N_z = 512$ it amounts to about 1 second and 8 seconds, respectively, on a standard laptop computer. Our algorithm also require pairs of fast direct and inverse Fourier transforms with complexity scaling as $\mathcal{O}(N_z \log N_z)$ that have a negligible impact on the total computation time.

## C  Examples of peculiar non-stationary states

In this last appendix we report two examples of initial states leading to a non-trivial dynamics that can be understood correctly only with the analysis of the Lax spectrum thanks to the soliton indicator.

We first consider the state of Eq. (9), with $\alpha = 2$ and a non-linearity $gn_0 = 2 \times 10^4$. Since the initial density dip is very narrow, we use a grid size of $N_z = 1024$ to correctly compute the time-evolution. In that case, the analytical result predicts that there is a single dark soliton at $\zeta_0 = -\frac{k_0}{4}$ in the gap. Our soliton indicator method detects $N_{sol} = 10 \pm 7$ solitons and a effective speed of sound $c_{eff} = 142.9 \pm 1.3$ very close to the bare speed of sound $c$. Figure 10a) and b) show the Lax spectrum structure, evidencing clearly the isolated eigenvalue corresponding to the dark soliton, and a bunch of discrete eigenvalues very close to the gap edges, that are within the uncertainty of our detection method.

Figure 10c) displays the density colormap of the time-evolution for this initial state. One can clearly see a dark soliton slowly moving due to the background phase gradient $k_0 \simeq \pi/L$, and several bright features propagating at a speed significantly faster than the bare speed of sound. We interpret this as a Bogoliubov excitation propagating at a group velocity $v_g = \frac{d\omega}{dk}$ larger than the speed of sound, which do not appear as a soliton in the Lax spectrum. We note that an analysis based only on the apparent speed of propagating features in Fig. 10c) would lead to an incorrect estimation of the speed of sound. We have checked that a initial state with a small Bogoliubov excitation wave-packet on top of a homogeneous background results indeed in a similar dynamics, except for the slow moving soliton that is absent.

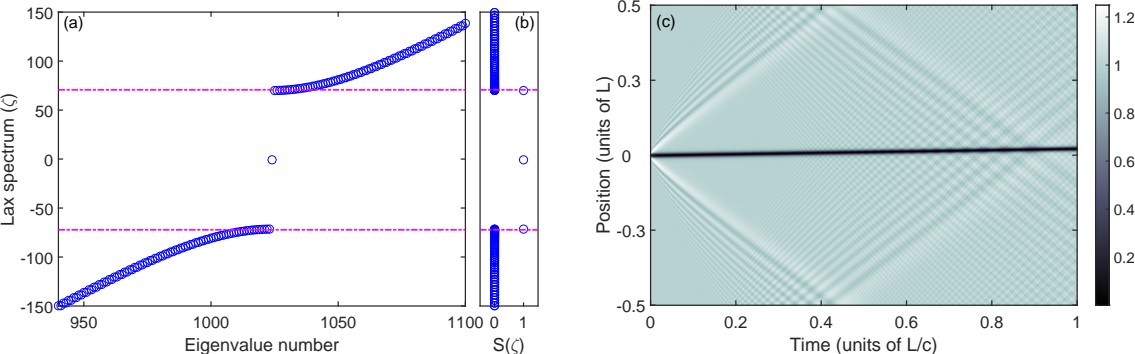

Figure 10: (color online) a) Lax spectrum as a function of the eigenvalue number for $\alpha = 2$ (open blue circles) using the initial state of Eq. (9). The two horizontal dashed-dotted magenta lines indicate the gap in the continuous spectrum, detected using the soliton indicator $S(\zeta)$. b) Soliton indicator $S(\zeta)$ for each eigenvalue, computed for the threshold $\epsilon_-$. c) Density colormap showing the propagation of the initial state.

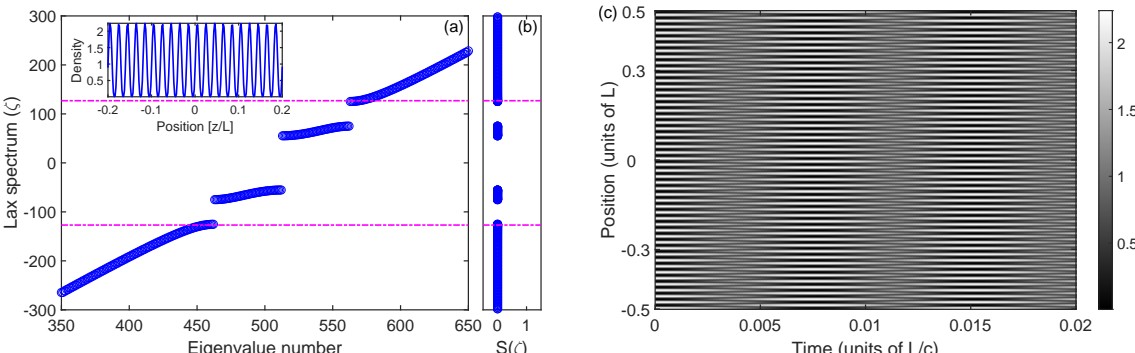

Figure 11: (color online) a) Lax spectrum as a function of eigenvalue number for a highly periodic initial state (open blue circles). Inset: initial density profile as a function of the position, zoomed in around $z = 0$. b) Soliton indicator $S(\zeta)$ for each eigenvalue, computed for the threshold $\epsilon_-$. In a) and b) the two horizontal dashed-dotted magenta lines indicate the edges of the main gap in the continuous spectrum. c) Density colormap of the periodic state for this particular realization (note the small time scale).

Finally we present a second puzzling case, showing that it is possible to construct a non stationary state with a non trivial Lax spectrum exhibiting several continuous branches, separated by multiple gaps. To do so we consider the initial state:

$$\psi_k(z) = \sqrt{n_0}\left(1 + \eta \cos kz\right),$$

with $n_0 = 1$, $\eta = 0.5$, $k = 50 \times 2\pi/L$ and nonlinearity $g n_0 = 2 \times 10^4$.

Figure 11 shows the Lax spectrum, the soliton indicator and the typical time evolution of the density profile, for this initial state. The Lax spectrum displays a non trivial structure, with many eigenvalues inside the gap. However, the analysis of our soliton indicator indicates that these eigenvalues do not correspond to solitons but rather behave as plane waves, in term of degeneracy. We interpret this as a spectrum with four continuous branches, separated by three gaps. Here since the soliton indicator is always 0 we define the gap based on the analysis of the distance between consecutive eigenvalues. This structure seems to arise from the highly periodic pattern of the initial state, that give rise also to a time periodic evolution at a high frequency. We also note that the density evolution do not display the propagation

of many gray solitons. We have carefully checked that this is not the result of a numerical artifact. Although we do not think that this state is of particular significance for the study of the physical properties of the 1DNLSE, its existence supports the need for a careful analysis of Lax spectrum to detect eigenvalues corresponding to solitons. We leave for a future work the question of the possibility of engineering a state with both continuous branches and soliton eigenvalues inside the gaps.

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
