# Peer review of "Characterizing far from equilibrium states of the one-dimensional nonlinear Schr{ö}dinger equation"

_SciPost Physics Core, doi:SciPost Phys. Core 8, 028 (2025)_

## Round 2 · Referee Report · Anonymous (Referee 1) · 2024-12-2

Strengths

1- The manuscript is based on a simple yet powerful idea 2- Well written and easy to follow 3- The proposed method is potentially useful for numerical simulations 4- As per the evidence provided the proposed method appears to work well for identifying dark and grey solitons

Weaknesses

1- Restrictive use cases - the proposed method is limited to homogeneous backgrounds where the inverse scattering theory applies 2- Not useful for experimental studies where usually only density information but not phase information is available

Report

This is a well-written manuscript about a new method to detect dark and grey solitons in numerical solutions of the nonlinear Schroedinger equation starting from a simple idea that is based on the powerful inverse scattering transform. Section 2 where the soliton indicator is introduced is very clear and easy to follow. The referencing of previous work appears adequate.

I was a bit confused by Sec. 3.1 where empirical thresholds $\epsilon_-$ and $\epsilon_+$ given by numerical values are introduced while no reference is made to the theoretically motivated threshold defined in Sec. 2, $\epsilon = \pi^2/(4L^2\sqrt{gn_0})$. As the argument in Sec. 2 appears to make perfect sense, why not use the numerical studies in Sec. 3 to validate it (or understand in which situations it may fail)?

While the proposed method appears to be potentially useful for numerical studies of generalized hydrodynamics, the introduction also references previous work where solitons had to be identified from experimental data (Ref. [46]). I'd like to encourage the authors to consider and comment on whether their approach could be extended to deal with missing information, i.e. specifically the situation where only density but no phase information is available.

Requested changes

1- Please add a discussion relating the empirical thresholds $\epsilon_-$ and $\epsilon_+$ to the theoretical threshold $\epsilon$ of Sec. 2. Moreover, it would be potentially more instructive and generally useful to give the thresholds in unit of $\epsilon$ rather than just as bare dimensionless numbers (which only make sense for particular chosen numerical parameters). 2- The blue solid line in Fig. 6a for eigenvalues identified by the threshold $\epsilon_+$ appears to zero for most bins. Does that mean that this threshold fails? Please discuss the implication, or correct the plot if this is just a mistake. 3- The same figure caption mentioned cyan lines corresponding to some some numerical factor multiplied to c. What is the signifcance of the particular factor? Why were these lines included in the plot? Please clarify!

Recommendation

Ask for minor revision

  • validity: high
  • significance: ok
  • originality: high
  • clarity: high
  • formatting: excellent
  • grammar: excellent

Author:  Romain Dubessy  on 2025-01-10  [id 5104]

(in reply to Report 1 on 2024-12-02)
Category:
answer to question

We provide a file 'reply.pdf' with a detailed reply to the remarks and comments of the anonymous Referee

Attachment:

reply.pdf

---

## Round 3 · Referee Report · Joachim Brand (Referee 1) · 2025-2-13

Report

The authors have addressed all of the questions that I had to my satisfaction and have revised the manuscript improving the clarity of the presentation.

I understand that the technique described in this paper will be useful for quantifying the number of solitons in wave systems where both amplitude and phase can be measured, as for example in water waves.

I support the publication of the current version of the manuscript as it easily meets the acceptance criteria of SciPost Physics Core.

Recommendation

Publish (easily meets expectations and criteria for this Journal; among top 50%)

---

## Round 3 · Author Response

Dear Editor,

Please find herewith enclosed a revised version of our manuscript entitled "Characterizing far from equilibrium states of the one-dimensional nonlinear Schrödinger equation".

We are grateful to the Referee for the careful reading of our manuscript. We believe that the Referee's remarks helped us to improve the clarity of the paper. We have taken into account all his/her comments and answered the questions he/she raised in the reply.pdf file, including a part highlighting the changes we made to the text.

We hope that our revised manuscript can now be considered for publication in SciPost Physics Core.

Yours Sincerely,
Abhik Kumar Saha and Romain Dubessy

---

## Round 3 · List of Changes

We have clarified the relation between the analytical threshold of section 2 and the two empirical thresholds of section 3.
We have made minor modifications to the figures and captions to take into account the Referee's remarks.

---

## Editorial Decision

published